# Synergistic Effects of Melatonin and Gamma-Aminobutyric Acid on Protection of Photosynthesis System in Response to Multiple Abiotic Stressors

**DOI:** 10.3390/cells10071631

**Published:** 2021-06-29

**Authors:** Aida Shomali, Sasan Aliniaeifard, Fardad Didaran, Mahmoud Lotfi, Mohammad Mohammadian, Mehdi Seif, Wacław Roman Strobel, Edyta Sierka, Hazem M. Kalaji

**Affiliations:** 1Photosynthesis Laboratory, Department of Horticulture, Aburaihan Campus, University of Tehran, Pakdasht 3391653755, Iran; aida.shomali@ut.ac.ir (A.S.); f.didaran@ut.ac.ir (F.D.); mlotfi@ut.ac.ir (M.L.); M.mohamadian.123@ut.ac.ir (M.M.); mehdi.seif@alumni.ut.ac.ir (M.S.); 2Institute of Technology and Life Sciences (ITP), Falenty, Al. Hrabska 3, 05-090 Raszyn, Poland; w.strobel@itp.edu.pl; 3Faculty of Natural Sciences, Institute of Biology, Biotechnology and Environmental Protection, University of Silesia in Katowice, 28 Jagiellonska, 40-032 Katowice, Poland; edyta.sierka@us.edu.pl; 4Department of Plant Physiology, Institute of Biology, Warsaw University of Life Sciences—SGGW, 02-787 Warsaw, Poland

**Keywords:** chemical priming, energy flux, OJIP, photosynthesis, *Vicia faba*

## Abstract

GABA (gamma-aminobutyric acid) and melatonin are endogenous compounds that enhance plant responses to abiotic stresses. The response of *Vicia faba* to different stressors (salinity (NaCl), poly ethylene glycol (PEG), and sulfur dioxide (SO_2_)) was studied after priming with sole application of GABA and melatonin or their co-application (GABA + melatonin). Both melatonin and GABA and their co-application increased leaf area, number of flowers, shoot dry and fresh weight, and total biomass. Plants treated with GABA, melatonin, and GABA + melatonin developed larger stomata with wider aperture compared to the stomata of control plants. The functionality of the photosynthetic system was improved in primed plants. To investigate the photosynthetic functionality in details, the leaf samples of primed plants were exposed to different stressors, including SO_2_, PEG, and NaCl. The maximum quantum yield of photosystem II (PS II) was higher in the leaf samples of primed plants, while the non-photochemical quenching (NPQ) of primed plants was decreased when leaf samples were exposed to the stressors. Correlation analysis showed the association of initial PI_abs_ with post-stress F_V_/F_M_ and NPQ. Stressors attenuated the association of initial PI_abs_ with both F_V_/F_M_ and NPQ, while priming plants with GABA, melatonin, or GABA + melatonin minimized the effect of stressors by attenuating these correlations. In conclusion, priming plants with both GABA and melatonin improved growth and photosynthetic performance of *Vicia faba* and mitigated the effects of abiotic stressors on the photosynthetic performance.

## 1. Introduction

Plants respond endogenously to abiotic stresses at the molecular and cellular levels. Understanding the underlying mechanism for stress tolerance is essential to develop stress-tolerant crops) [1]. Experiencing single stressor enhances subsequently plant tolerance to diverse abiotic stresses, which is known as cross-stress tolerance [2,3]. Therefore, the mechanisms involved in response to diverse abiotic stresses have sharing similarities in some pathways or signaling molecules. Various approaches have been employed for enhancing plant tolerance to abiotic stresses; however, some of those approaches are tedious (e.g., conventional breeding), and some are controversial (e.g., plant genetic modification) [4]. Considering these limitations, attentions are directed toward priming of plants to improve their stress tolerance. Priming of plants can be initiated naturally when non-lethal stresses come about, that is, associated with the promotion of tolerance mechanisms [5,6,7], which diminishes the negative impact of stress factor on growth and physiological responses of plants. Both biotic and abiotic agents have been used for priming of plants. Although biotic priming, using plant growth-promoting bacteria, has been used to increase crop tolerance to different stressors [8,9,10], chemical priming attracted a great deal of attention due to its ease of application [4]. It has been reported that various types of chemical compounds can act as priming agent against diverse abiotic stresses [11]. Some example of those are phytohormones such as polyamines [12] and salicylic acid [13,14], mild stress pre-exposure such as different salts [13] and reactive oxygen species [3].

Initial priming in plants is regulated through different mechanisms, including synthesis of endogenous compounds to enable plant coping with stresses mostly through boosting protecting mechanisms on photosynthesis machinery [15,16].

Melatonin (*N*-acetyl-5-methoxytryptamine) and GABA (γ-aminobutyric acid) are two endogenous molecules that take part in regulation of plant responses to environmental variables. Melatonin is a pleiotropic molecule with multiple cellular and physiological functions. The discovery of melatonin in plants occurred in 1995 [17,18]. GABA (γ-aminobutyric acid) is a non-protein, four-carbon amino acid that exists in the plant kingdom and was discovered in plants more than half a century ago [19]. It has been reported that both GABA and melatonin enhance photosynthesis performance of plants [9,20,21,22,23,24]. For instance, exogenous application of 20 mM GABA on *Capsicum annuum* L. enhanced its photosynthetic capacity by increasing maximum quantum yield of photosystemII (PSII) (F_V_/F_M_), actual PSII photochemical efficiency (ΦPSII), and electron transport rates [22]. The same effects of GABA on improving photochemical efficiency of photosynthesis as well as the role GABA plays in reduction of non-photochemical quenching (NPQ) have been reported in muskmelon especially under hypoxia stress [25]. Promotion of photosynthesis, chlorophyll content, growth, and biomass was also achieved in maize plants after exogenous application of 50 mg L^−1^ GABA [23]. In the same plant, Seifikalhor et al. (2020) reported improving effects of 25 µM and 50 µM of GABA on photosynthesis and vegetative growth. In their study, 50 µM of GABA was attributed to the highest biomass production [9].

Plants hydroprimed with melatonin were characterized with higher plant growth and yield [26]. Testing different concentrations of melatonin (10, 20, 30, 40, or 50 µM) on *Brassica juncea* revealed a dose-dependent increase in plant growth, photosynthesis, and antioxidant activities [27]. The protecting effects of exogenous melatonin on the photosynthetic apparatus have been reported on *Chara australis*. In this plant, exogenous application of 10 µM of melatonin resulted in better functionality of the photosynthetic electron transport system. The increased quantum yield of PS II due to melatonin application was postulated to be the result of increased open reaction centers of PSII rather than improved efficiency of each reaction center [21]. In *Pisum sativum* L., hydropriming by melatonin (50 or 200 µM) resulted in increased water splitting efficiency at the donor site of PSII (F_V_/F_O_), F_V_/F_M_, and chlorophyll fluorescence decrease ratio.

Since plants normally increase their endogenous melatonin [26] and GABA [28,29] levels in response to stress conditions, many attempts have been made to simulate those endogenous responses through their exogenous applications. For instance, exogenous GABA has been used to alleviate negative effects of various abiotic stresses, such as salt stress [30,31], light stress [22], cadmium stress [9], drought stress [16], etc. Similarly, melatonin has been applied to minimize the adverse effects of various abiotic stresses, such as drought [32], high-light stress, temperature stress [33], oxidative stress [34], and nitrate toxicity [35].

A number of evidences suggest that melatonin play its role in stress response by inducing changes in membrane and protein peroxidation [36,37], through its antioxidant function [38] or by regulating circadian clock responses [39].

There are some documents advocating the interactive effects of GABA and melatonin. For instance, it has been reported that melatonin triggers GABA-shunt during stress responses [40]. An increasing number of studies focused on phytochemical and biochemical effects of melatonin and GABA on improving plant performance and alleviation of the stress impacts on plants, while the effect of melatonin and GABA on biophysical aspects of photosynthesis in normal and stress condition have been poorly investigated. Therefore, in this study, the possibility of using GABA (20 µM), melatonin (200 µM), and GABA + melatonin (GABA (20 µM) + melatonin (200 µM)) as priming agents for improving plants tolerance (by focus on photosynthesis functionality) to different stressors (PEG, salinity (NaCl), and sulfur dioxide (SO_2_) was investigated using the faba bean as a model plant for stomata and photosynthesis studies. Although concerns regarding drought and salinity has been accentuated previously, increase in the concentration of SO_2_ in the atmosphere due to the recent increase in rate of industrialization [41] and the adverse effects of SO_2_ on plants has also emerged as a matter of concern [42,43]; thus, the possibility of alleviation of SO_2_ stress on plants by melatonin and GABA, which has not been reported yet, was also targeted in the present study. This study can also be distinguished from previous studies since it targets both sole and combined effects of GABA and melatonin on plant growth and stress responses.

## 2. Materials and Methods

### 2.1. Plant Material and Growth Conditions

*Vicia faba* seeds were sown in pots with 15 cm diameters filled with mixture of coco-peat and perlite (2:1 *v*/*v*). Plants were grown under greenhouse conditions, and after seedling emergence, they were irrigated with modified Hoagland solution (half strength) three times per week. Ten days following seedling emergence, full-strength Hoagland solution was used for irrigation of the plants. GABA (20 µM), melatonin (200 µM), and GABA (20 µM) + melatonin (200 µM) were applied through irrigation once per week when the first true leaves emerged on the plants.

### 2.2. Measurements of Morphological Traits

Growth and morphological characteristics, such as root volume, fresh and dry weights, shoot fresh and dry weights, stem diameter, specific leaf area (SLA), number of leaves, and number of flowers were measured 45 days following seedling emergence. Flowering times were recorded when the flower primordia emerged on the plants. For measurement of SLA, first-leaf area was calculated by scanning of the detached leaves; then, their leaf area was calculated using Digimizer software (Digimizer V 4.1.1.0). To measure the dry weight, the samples were dried in an air-forced oven for 72 h at 75 °C. SLA was calculated by the ratio between leaf area and its dry weight. Root volume was measured based on the method described by [44].

### 2.3. Chlorophyll Index

A SPAD meter (Model: SPAD-502; Konica Minolta Sensing, Osaka, Japan) was used to directly estimate the chlorophyll content of the leaves. In each plot, the uppermost fully developed leaves were used for the SPAD readings. Since the sampling sensor area of the SPAD meter is small, the measurements were taken from three different spots of each leaf, for representative measurements, the average was used as the SPAD readings [45]. Finally, the SPAD meter readings were converted to actual chlorophyll content using the following conversion relationship where CC is chlorophyll content (μmol m^−2^), and SMR is the SPAD meter reading.
(1)$$\mathrm{CC}=10^{\left(SMR^{0.265}\right)}$$

### 2.4. Stomatal Morphology and Conductivity

Stomatal morphology (stomatal length, stomatal width, pore aperture, stomatal index, and density) were measured on the young, fully developed leaves from each treatments according to the method described by [44]. A section of the young, fully developed leaf located midway between the tip and the base and equally distanced from each edge was used for microscopic analysis. To prepare the samples, the lower epidermis was coated with a thin layer of nail polish. After 10 min, a strip of transparent sticky tape was used to remove the dried polish. The dry polish sample along with the adhered sticky tape mounted on microscope slides and stomatal morphological details were investigated under a light microscope. Images were taken by an Omax top view software (version 3.5) and further analyzed using imageJ software (U.S. National Institutes of Health, Bethesda, MD, USA; http://imageJ.nih.gov/ij/ Version 1.53j 13 May 2021) to record stomatal length, stomatal width, pore length, aperture, and stomatal density. In total, 100 stomata from the leaves of each group were analyzed.

Stomatal conductance (*g_s_*) was measured according to the method described by Fnourakis et al. (2013). In this calculation, stomatal pore depth was considered to be equal to the guard-cell width (i.e., stomatal width/2), assuming guard cells inflate to a circular cross-section. A 100× magnification was used to assess stomatal density [46,47]. Number of stomata was counted on three randomly chosen areas of the same leaf from which stomatal size measurements were taken. Analysis was done on 1 mm^2^ of the middle of the leaf on both sides of the main vein [48]. Calculation of *g_s_* was done based on the following equation:(2)$$\left[gs=\frac{\left(diffusion\;coefficient\right)\times \left(stomatal\;density\right)\times \left(\pi \times poreapperture÷2\times porelength\;÷2\right)}{\left(molar\;volume\;of\;air\right)\times \left[\;\left(pore\;depth\right)+\sqrt{\left(pore\;apperture\;÷2\times pore\;length\;÷2\right)}\right]}\right]$$

### 2.5. Chlorophyll Fluorescence Quenching Analysis

Samples were taken 45 days after growth from fully developed mature leaves. The samples were dark-adapted for 20 min after treating with PEG, NaCl, and SO_2_ and were immediately used to measure slow induction of chlorophyll fluorescence with a FluorCam (FluorCam FC 1000-H, Photon Systems Instruments, PSI, Drasov, Czech Republic). A CCD camera and four fixed LED panels were included in the FluorCam. LED panels were used for supplying measuring pulses and for induction of saturating flash (https://fluorcams.psi.cz/products/handy-fluorcam/#details). F_V_/F_M_ was calculated using a custom-made protocol [48]. Measurements of chlorophyll fluorescence were started during exposure of the samples to short flashes in darkness and following a saturating pulse (3900 μmol m^−2^ s^−1^ Photosynthetic photon flux density (PPFD) at the end of the measurement to stop the electron transport as a result of reduction of quinone acceptors [49]. Based on the protocol, two sets of fluorescence data were recorded: one averaged over the time of short flashes in the darkness (F_o_) and the other at the time of exposure to saturating flash (F_M_).

For determination of NPQ, maximum fluorescence in light-adapted steady state (F_M_′) was recorded with that used for calculation of NPQ. The related data and calculations were done using version 7 (PSI, Drasov, Czech Republic) of FluorCam software [50].

The maximum quantum yield of PSII (F_v_/F_M_) was calculated using the following equation:F_v_/F_M_ = (F_M_ − F_o_)/F_M_(3)

NPQ was calculated based on the following equation:NPQ = (F_M_/F_M_′) − 1(4)

The photochemical quenching or the amount of open PSII reaction centers at every consecutive pulse remains only influenced by the adaptation to the available light intensity and environmental conditions. Besides, two additional saturating pulses (F_E_″ and F_T_″ respectively) were given in darkness after completing the diurnal light cycle. This technique allows approximating the relaxation kinetics of the NPQ mechanisms, disentangling the energy-dependent quenching (qE), state-transition quenching (qT), and photo-inhibitory quenching (qI) based on their relaxation kinetics [51,52] according to the following equations:(5)$$\mathrm{qE}=\frac{{F^{\prime \prime }}_{E}-{\mathrm{Fm}}^{\prime }}{{\mathrm{Fm}}^{\prime }}\;$$
(6)$$\mathrm{qT}=\;\frac{{F^{\prime \prime }}_{T}-{F^{\prime \prime }}_{E}}{{\mathrm{Fm}}^{\prime }}\;$$
(7)$$\mathrm{qI}=\;\frac{F_{m}-{F^{\prime \prime }}_{T}}{{\mathrm{Fm}}^{\prime }}\;$$

### 2.6. Polyphasic Chlorophyll Fluorescence (OJIP) Transients

OJIP protocol was performed on fully developed, mature leaves of plants after 45 days of growth. The samples were dark-adapted for 20 min. A Fluorpen FP 100-MAX (Photon Systems Instruments, Drasov, Czech Republic) was used for measuring the OJIP transients. Different biophysical and phenomenological parameters related to PSII status [53] was investigated by JIP-test according to the protocol described by [20]. Parameters derived from the OJIP protocol provides information on the energy fluxes of light absorption (ABS) and trapping (TR) of the excitation energy and electron transport (ET_O_) per reaction center (RC), which are described on the Appendix A.

### 2.7. Application of Abiotic Stresses

#### SO_2_ Stress

To impose SO_2_ stress, plant samples were prepared based on the previously described protocols [44,48,54]. Leaf samples were cut from the petiole under water in order to prevent stress on the leaves during the analysis; each leaf was placed in 2 mL vials containing distilled water. The vials were placed into a sealed glass chamber (20 × 20 × 5 cm) with constant PPFD of 300 μmol m^−2^ s^−1^ at 25 ℃ and incubated for 6 h with different concentrations of SO_2_ (0, 0.5, 1, 2 ppm) before further analysis. Pure SO_2_ was provided from a 5-L capsule with 99.99% purity. To calculate the inlet gas to the containers, two computational methods were used, including the ideal gas low [55] and Van der Waals equation [56]. All experimental procedures were conducted at 25 ℃ and air pressure of 1 Pa.

### 2.8. Salinity and Osmotic Stresses

PEG 8000 (0 and −8 bar) and NaCl (0 and 100 mM) were used to impose salt and osmotic stresses on the leaf discs of primed (GABA, melatonin, and GABA+ melatonin) plants. Leaf disks were cut from 60-day-old faba bean plants. Seven leaf disks were put into distilled water (control), NaCl, and PEG solutions, then placed in a growth chamber with constant light (300 μmol m^−2^ s^−1^ PPFD) at 25 ℃ and incubated for 6 h before chlorophyll fluorescence analysis [44,54].

### 2.9. Statistical Analysis

The results represented the average mean values of four replications. The data were analyzed using SAS software (version 9.0). The two-way analysis of variance (ANOVA) was used to find the significant differences (*p* ≤ 0.05), and then the Duncan multiple comparisons test was used to compare the means. For analyzing chlorophyll fluorescence parameters, obtained data were subjected to two-way ANOVA, and for mean comparison, the Tukey multiple comparison test was used. For stomatal characteristics, data obtained from one leaf were considered not independent, and for mean comparison, one-way ANOVA as well as Tukey multiple comparison test was used.

## 3. Results

### 3.1. GABA and Melatonin Synergistically Improve Growth

The concentration of GABA and melatonin was selected according to the optimum concentration of GABA and melatonin that were reported to have a positive effect on plant abiotic stress tolerance [57,58,59]. Growth traits were significantly (*p* > 0.01) affected by priming treatments. The widest leaf area was obtained in plants treated with GABA + melatonin, and the lowest leaf area was observed in control plants. Plants treated with GABA + melatonin, GABA, and melatonin, respectively, showed 113%, 39%, and 38% increase in leaf area when compared to the leaf area of control plants (Figure 1).

Root volume was increased in plants treated with GABA + melatonin and melatonin; however, GABA did not significantly affect root volume of faba bean plants. The largest root volume was obtained in plants treated with GABA + melatonin, by 23% larger root volume than the control (Figure 2a).

The maximum number of flowers was observed in GABA and GABA + melatonin, respectively, by 56% and 55% increase in comparison with the number of flowers in the control. This increase in the number of flowers by melatonin was 39% compared with control (Figure 2b).

Plants treated with GABA + melatonin exhibited the highest fresh and dry weights of shoots, which showed 55% increase in shoot dry weight compared to the control. GABA and melatonin also led, respectively, to 30% and 34% increase in dry weight of shoot compared to the control (Figure 2c,d).

GABA+ melatonin treated plants also showed 37% increase in total biomass compared to the control. Total biomass of plants treated with either of GABA or melatonin was not significantly different from the control (Figure 2d).

### 3.2. Generation of Large Stomata and Improved Stomatal Conductance by GABA and Melatonin

Stomatal characteristics were significantly (*p* > 0.01) affected by the GABA and melatonin treatments (Figure 3a–d and Figure 4). The highest stomatal length was obtained in plants treated with GABA + melatonin, which showed 7% longer stomata than the control. Moreover, GABA and melatonin caused 3% and 4% increase in stomatal length in comparison with the stomatal length of control (Figure 3a). Plants treated with GABA + melatonin had the widest stomata, showing 3.5% increase in stomatal width than the control. The increase in width of stomata was 2.4% and 2.5% by sole application of GABA and melatonin, respectively (Figure 3b). The maximum length of stomatal pore was also observed in plants treated with GABA + melatonin, which showed 9.5% longer stomatal pore than the control. GABA- and melatonin-treated plants showed, respectively, 5.6% and 4.7% increase in length of their stomatal pore in comparison with pore length of control plants (Figure 3c). Stomatal aperture (pore width) in plants treated with GABA + melatonin was three-fold wider than the stomata aperture of the control plants. GABA- and melatonin-treated plants also showed 10.1% and 4.5% wider aperture in comparison with stomatal aperture of the control (Figure 3d and Figure 4).

*g_s_* was also increased in primed plants compared to control. The *g_s_* in plants primed with GABA + melatonin, melatonin, and GABA was, respectively, 2.1-, 1.5-, and 1.3-fold higher than the *g_s_* of control plants (Figure 3e).

### 3.3. GABA and Melatonin Synergistically Improve Photosynthetic Functionality

Chlorophyll content in plants treated with GABA + melatonin, GABA, and melatonin showed, respectively, 72%, 61%, and 60% increase in comparison with the chlorophyll content in the leaves of control plants (Figure 5).

The OJIP transient showed significant difference (*p* > 0.01) among treatments (Table 1). The maximum value of F_V_/F_M_ was obtained in plants treated with GABA + melatonin, which showed 4.8% increase compared to the F_V_/F_M_ of control plants (Table 1). GABA- and melatonin-treated plants also exhibited, respectively, 3.7% and 3.3% increase in F_V_/F_M_ in comparison with the control. Plants treated with GABA + melatonin, GABA, and melatonin had, respectively, 2.2-, 2-, and 1.8-fold higher PI_abs_ than the control plants (Table 1).

ABS/RC was reduced by about 15% through priming treatments compared to ABS/RC of control plants. No significant difference was observed among ABS/RC of the primed plants (Table 1).

ET_O_/RC increased, respectively, by about 10%, 13%, and 6% in plants primed with GABA, GABA + melatonin, and melatonin when compared with the ET_O_/RC in the control plants (Table 1). Although no significant difference was observed among TR_O_/RC of treated plants, they had approximately 11% lower TR_O_/RC than the control plants (Table 1). DI_O_/RC reduced by about 2.5% in treated plants in comparison with the control; however, no significant difference was observed among DI_O_/RC of primed plants (Table 1, Appendix A).

### 3.4. GABA and Melatonin Priming Synergistically Improve Tolerance to Abiotic Stresses

#### 3.4.1. SO_2_ Toxicity

Exposing leaf samples to SO_2_ resulted in significant difference (*p* > 0.01) among F_V_/F_M_ of dark-adapted samples of plants that were primed with GABA, melatonin, and GABA + melatonin. Although exposure to SO_2_ decreased F_V_/F_M_ in control samples, this decrease was not detected or was smaller in primed samples (Figure 6a). Lowest F_V_/F_M_, among all samples was detected in those exposed to 2 ppm SO_2_. Following exposure to different concentrations of SO_2_, F_V_/F_M_ in GABA + melatonin samples was higher than control. F_V_/F_M_ in GABA + melatonin primed samples, in response to 0.5, 1, and 2 ppm SO_2_ was 22%, 16%, and 22% higher than the F_V_/F_M_ of control, respectively. The minimum value of F_V_/F_M_, regardless of SO_2_ concentrations was associated with the control samples (Figure 6a and Figure 7).

The value of NPQ in leaf discs of primed plants was significantly (*p* > 0.01) influenced by both priming (GABA, melatonin, and GABA + melatonin) and SO_2_ concentration. In control samples, by exposure to 2 ppm SO_2_, a drastic reduction detected in NPQ compared to 0 ppm SO_2_, while in samples of plants primed with GABA + melatonin, a marked increase in NPQ was detected by exposure to 2 ppm SO_2_ (Figure 6b and Figure 8). Samples of plants primed with GABA also demonstrate the same reduction in NPQ as that of the control, and no significant difference was observed among samples of plants primed with melatonin in different concentrations of SO_2_. In general, the maximum NPQ was detected in samples of plants primed with GABA + melatonin when exposed to 2 ppm SO_2_, which showed 47% increase in comparison with the same priming samples exposed to 0 ppm SO2 (Figure 6b and Figure 8, Appendix A). The components of NPQ were also affected by SO_2_ stress and priming agents. Highest qE was obtained in plants primed with melatonin, and priming with GABA + melatonin resulted in the highest qT and qI (Appendix A).

#### 3.4.2. Osmotic Stress

Exposing dark-adapted leaf discs of primed plants to PEG resulted in significant differences (*p* > 0.01) among F_V_/F_M_ of primed samples (Figure 6c). Under non-osmotic stress condition, the maximum F_V_/F_M_ was obtained in plants primed with GABA + melatonin and melatonin, showing, respectively, 8% and 7% higher F_V_/F_M_ compared to F_V_/F_M_ of control samples.

The maximum NPQ in samples that were exposed to PEG was detected in leaf samples of plants primed with melatonin and GABA, which showed 8% and 10% higher NPQ than the NPQ of control (Figure 6d and Figure 9). In general, NPQ was increased in the samples of plants primed with GABA and melatonin, while in leaf samples of control plants, a drastic reduction in NPQ was detected (Figure 6d and Figure 9; Appendix A). The priming agent also affected NPQ components of plants under osmotic stress. Priming by melatonin reduced qE and qI components of NPQ, which resulted in higher qT than the other components in plants primed with GABA, qT was also highest among NPQ components (Appendix A). For GABA + melatonin, all NPQ components contributed equally.

#### 3.4.3. NaCl Stress

Exposing primed samples to 100 mM NaCl resulted in significant differences (*p* > 0.01) among F_V_/F_M_ of dark-adapted samples (Figure 6e and Figure 10). After exposing leaf samples to NaCl, the maximum F_V_/F_M_ was related to samples of plants that were primed with GABA + melatonin, which showed 16% higher F_V_/F_M_ than salt-exposed samples of control plants. After exposure to 100 mM NaCl, F_V_/F_M_ of samples obtained from plants that were primed with sole GABA and melatonin was also higher than the F_V_/F_M_ of control, which showed 15% higher F_V_/F_M_ than the control samples. In general, under NaCl condition, primed plants with GABA, melatonin, and GABA + melatonin showed no significant difference with the F_V_/F_M_ of their control, while NaCl exposure to control samples caused a significant decrease compared to the non-stressed control samples (Figure 6e and Figure 10).

NPQ was increased in response to NaCl stress in leaf disks of plants primed with GABA and melatonin, while the value of NPQ in samples of control was reduced by exposure to 100 mM NaCl (Figure 6f and Figure 10). The maximum value of NPQ was detected in samples of plants that were primed with GABA and exposed to 100 mM NaCl, which showed 24% higher NPQ compared to the control samples. Moreover, NPQ in leaf samples of plants primed with GABA and melatonin was, respectively, 24% and 14% higher than the control samples when exposed to salt stress (Figure 6f and Figure 10, Appendix A).

Exposing primed plants to NaCl stress caused more contribution of qT and qI to the NPQ than the qE in GABA-primed samples. In plants primed with melatonin, qI was lower than qE and qT. The components of NPQ showed almost the same value in plants primed with GABA + melatonin.

### 3.5. GABA and Melatonin Affect the Correlation of Initial PI_abs_ with F_V_/F_M_ and NPQ after Exposure to Stress

To better understand the association between the initial photosynthesis efficiency of plants and their ability to tolerate abiotic stresses, a correlation analysis between the initial PI_abs_ (before exposure to stressors) and quenching parameters of leaf samples of primed plants exposed to stressors was conducted. The level of correlation among those traits was differed in primed plants and was affected by stressors.

The initial PI_abs_ and F_V_/F_M_ showed −98% correlation in non-primed plants when the leaf samples were exposed to PEG stress. While in leaf samples of plants primed with GABA, the negative correlation was decreased to −19%, and leaf samples of plants primed with melatonin and GABA + melatonin showed positive correlations (57% and 31%, respectively). Under NaCl stress, leaf samples of plants had −70% correlation between initial PI_abs_ and F_V_/F_M_, while in plants primed with GABA, melatonin, and GABA + melatonin, −16%, −15%, and 35% correlation, respectively, were detected. In absence of stressors, the correlation was −13%, 6%, 10%, and 28% in leaf samples of control, plants primed with GABA, melatonin, and GABA + melatonin, respectively. Under SO_2_ stress, the correlation was −80% in control plants, while, when leaf discs of plants primed with GABA were exposed to SO_2_, the correlation was reduced to −18% (Figure 11).

Contrasting differences were detected in correlation of the initial PI_abs_ and NPQ between primed and control plants under control or stress condition. The initial PI_abs_ and NPQ of leaf samples exposed to PEG showed −68% correlation in control; however, those traits were negatively correlated in plants primed with GABA, melatonin, and GABA + melatonin, which showed −3%, 81%, and −49% correlations, respectively. When leaf samples of primed plants were exposed to NaCl stress, −80% correlation was detected in the control condition, while 79%, 36%, and 22% correlation was detected in plants primed with GABA, GABA + melatonin, and melatonin, respectively. In absence of stressors, PI_abs_ and NPQ were weakly correlated. Under SO_2_ stress, this correlation was −9% in the control condition, while more strong negative correlation was detected in leaf samples primed with GABA, melatonin, and GABA + melatonin, which showed −13%, −61%, and −83% correlation, respectively (Figure 11).

## 4. Discussion

Based on the finding of present study, melatonin and GABA increased leaf area of *Vicia faba*. The improving effect of GABA on leaf area has been also reported in *Vinga mungo* L. [60]. However, the negative effect of melatonin on leaf area has been reported [61,62]. Interestingly, when both GABA and melatonin were co-applied, the increase in leaf area was markedly higher than the increase obtained by sole application of either of GABA or melatonin. This result suggests an interactive effect for GABA and melatonin. In accordance, enhancing the effect of melatonin on GABA homeostasis through triggering GABA-shunt pathway has been reported [40].

Root volume was increased in plants treated with GABA + melatonin and melatonin; however, GABA did not significantly affect root volume of faba bean plants. Similarly, in muskmelon, GABA had no effect on root growth [63], which is in line with the findings of this study. The role of melatonin on de novo root formation in *Hypericum perforatum* L. [64], root elongation in rice [65], and promotion of adventitious and lateral root regeneration in barley has been also reported [66].

GABA application caused generation of more flowers in faba bean plants, which agrees with the previous reports on flowering enhancement in bitter gourd and white gourd [67,68]. The number of flowers of faba bean also increased in plants treated with melatonin, which can be related to the role of melatonin in the transition to flowering as reported in *Chenopodium rubrum* [39].

Dry and fresh weights of shoots were also increased in plants that were treated with GABA. The same finding was reported on shoot fresh weight of maize in response to exogenous GABA [9]. Moreover, the increase in dry weight of shoot obtained in this study is in line with the result of a previous study that showed melatonin-treated maize plants show enhanced fresh shoot weight under salt stress [69]. However, melatonin was shown to have no effect on dry weight of shoots in maize seedlings in normal environmental conditions [62].

In the present study, higher biomass production was obtained by co-application of GABA and melatonin compared to the biomass of the control. It was reported that GABA increased biomass of maize plants under cadmium stress [9]. In addition, increase in biomass production was reported in in vitro-regenerated plants of *Prunella vulgaris* L. in cultured media that were supplemented with melatonin [70] and in soybean plants developed from seeds that were primed with melatonin [17,71]. On the other hand, decreased biomass production due to melatonin application has been reported in cherry rootstock [71].

Despite of no significant effect of sole application of GABA or melatonin on biomass production in faba beans, co-application of GABA and melatonin resulted in the highest biomass production. This may be due to the accumulative effects of melatonin and GABA due to the involvement of melatonin in mediation of GABA-shunt [40] or inductive effect of melatonin on endogenous GABA [72].

Stomatal movements controls gas exchange, especially CO_2_ uptake for photosynthesis and water loss through transpiration [73]. Thereby, stomata play a crucial role in water use efficiency and productivity of plants [74]. Discovering the first phytomelatonin receptor (CAND2/PMTR1) in *Arabidopsis thaliana* revealed that melatonin takes part in the receptor-dependent stomatal closure [75]. It was demonstrated that the effect of melatonin on stomatal aperture was concentration-dependent in both *Arabidopsis thaliana* and *Vicia faba*, and increase in concentration of melatonin from 0 to 20 µmol/L reduced the aperture of stomata, while increase in melatonin concentration from 20 to 200 µmol/L promoted stomatal opening [75]. Meng et al. (2014), also demonstrated that osmotic stress reduced stomatal aperture, while melatonin pretreatment resulted in wider and longer stomata with bigger aperture [76]. This is in line with the result of this study that both width and length of stomata and stomatal pore increased in plants that were treated with melatonin. Beside stomatal aperture, melatonin also improved *g_s_*, suggesting a role for melatonin in regulation of stomatal gas exchange in *Vicia faba*. The same effect of melatonin on *g_s_* was also reported in cucumber and wheat by exogenous application of melatonin [32,77]. Increased stomatal aperture has both advantage and disadvantage for plant stress tolerance. In one hand, it increases the water loss from the plant; if the lost water is not compensated by more water uptake, it would be harmful for the plant [73]; on the other hand, CO_2_ restriction is a critical challenge for the plant photosynthesis under water stress circumstances. Therefore, stomatal opening would improve gas exchange efficiency; as a result, this improves stress tolerance by maintaining a steady state photosynthesis [77]. It has been reported that melatonin enhances osmotic stress tolerance by promoting the efficiency of CO_2_ assimilation [78] and obstructing the inhibitory effects of osmotic stress on *g_s_* [79]. Melatonin takes part in increasing *g_s_* by maintaining cell turgor that facilitates the stomatal opening and conductance [76]. Increased stomatal conductance aids better water and CO_2_ mobility and improves photosynthesis of melatonin-treated plants [77]. Moreover, melatonin down-regulates the key gene in ABA biosynthetic pathway (NCED3) and up-regulate the genes involved in degradation of ABA (CYP707A1 and CYP707A1), which improves the re-opening function of stomata and *g_s_* [80]. Moreover, it has been reported that regulation of water balance in *Malus* species is accompanied by up-regulation of melatonin synthesis genes (TDC1, AANAT2, T5H4, and ASMT1), suggesting a pivotal role of melatonin in plants’ water balance under abiotic stresses [23,80].

Interestingly, in plants that were treated with GABA, open stomata were along with improved electron transport system of photosynthesis, suggesting that GABA increases stomatal aperture to facilitate CO_2_ uptake to keep pace with improved light reactions of photosynthesis.

GABA exerted the same effect as melatonin on stomatal pore aperture. In plants primed with both GABA and melatonin, the stomatal aperture was also wider than the plants that were treated with either of GABA or melatonin, suggesting a synergistic effect of these two compounds on stomatal morphology.

Mekonnen et al. (2016) indicated the role of GABA in stomatal aperture by studying GABA-depleted *gad1/2* mutants of *A. thaliana* phenotype. The *gad1/2* mutant showed wilting phenotype under drought stress, which was due to increased stomatal aperture and disturbance in stomatal closure [81]. A major pathway-mediating ion transfer during stomatal opening (ALMT9) is inhibited by GABA under osmotic stress to improve osmotic stress tolerance, while in wild-type plants under normal condition, stomatal aperture in plants treated with GABA was higher than the control, and the increase in stomatal aperture was contributed to higher *g_s_* [82], which agrees with the result of our study on improved *g_s_* in GABA-primed plants. Furthermore, our result is in line with the previous reports on the effect of GABA on increasing *g_s_* in maize and rice [80,83]. It can be postulated that GABA improves stress tolerance by maintaining cell turgor, which improves *g_s_* and further enhancement of photosynthesis [22].

Chlorophyll metabolism is largely affected by environmental cues, which influences plant growth and yields [51]. Increase in chlorophyll content is shown to be an early response to environmental fluctuations [84]. Exogenous melatonin was shown to play a protective role by maintaining higher levels of chlorophyll in *Chara australis* [21]. Application of melatonin maintained higher content of chlorophyll during leaf senescence in barley [85] and alleviated chlorophyll degradation during salt [29], cold [86], and heat [87] stresses. Our data confirmed this effect, as chlorophyll content was increased in plants treated with melatonin and GABA, which is in agreement with the results obtained in black cumin [88]. Moreover, chlorophyll content in plants that were treated with GABA + melatonin was more than that of plants treated with either of GABA or melatonin. This suggests an interactive effect between GABA and melatonin on chlorophyll homeostasis.

When plants are exposed to light, the energy of light is absorbed by chlorophyll molecules in leaves and converted to different levels of excitation. Most of the environmental factors exert direct or indirect effects on light reactions of photosynthesis, and abiotic stresses alter the bioenergic balance of photosynthesis [89]; this effect can be detected by fluorescence of chlorophyll molecules [9]. To further explore the behavior of PSII of faba beans in response to GABA and melatonin, in the present study, imaging of chlorophyll a fluorescence quenching and JIP-test were applied. F_V_/F_M_ is a parameter derived from chlorophyll fluorescence data that reflects the maximum photochemical quantum yield of PSII and thus efficiency of photosynthesis [90]). In this research, both GABA and melatonin increased F_V_/F_M_ in faba beans. The increase in F_V_/F_M_ in response to exogenous GABA has been also reported in maize seedlings [9]. In addition, similar to our finding, melatonin also increased the value of F_V_/F_M_ in *Pisum sativum* L. [26].

ET_o_/RC, TR_o_/RC, and ABS/RC, which are parameters of specific energy fluxes, changed in response to the priming treatments. TR_o_/RC and ABS/RC decreased in primed plants, while ET_o_/RC was higher in primed plants compared to the control. Our result demonstrated that melatonin decreased ABS/RC. In line with our finding, reduction in ABS/RC has been reported in *Chara australis* following melatonin application [21]. Influence of GABA on specific energy fluxes in tomato, and consequently, improved ET_o_/RC has been reported [91].

PI_abs_ is a delicate parameter of JIP-test that indicates the photochemical performance of plants. This parameter is a multi-parametric expression of light energy absorption, excitation energy trapping, and conversion of excitation energy to electron transport that depicts photosynthetic activity through a reaction center complex of PSII [92]. It has been found that exogenous melatonin increases the value of PI_abs_. Since PI_abs_ is a function of ABS/RC and other chlorophyll fluorescence parameters, variation in the value of PI_abs_ reversely corresponds with ABS/RC. The increase in PI_abs_ in response to exogenous melatonin was also reported in *Chara australis* [21].

Photosynthetic performance is a central indicator of plants’ fitness to their growth conditions. Most stresses, including osmotic stress, affect chlorophyll fluorescence of plants. Tracking and analyzing the chlorophyll fluorescence provides a reliable, rapid, and non-invasive indicator to investigate photosynthetic function under different environments [93].

Plants need to regulate the absorbed light energy to abate the damage imposed to photosynthesis aperture. If the excess light energy is absorbed by plants, not quenched by photoprotective mechanisms or photochemistry, it can result in formation of damaging reactive-oxygen species, singlet oxygen, or superoxide [94,95]. Therefore, to avoid cellular damage and to maintain photosynthesis, plants optimize the absorbed light by directing its energy path from photosynthesis to photoprotective mechanisms. The efficiency of photoprotective mechanisms can be also quantified by chlorophyll fluorescence. For instance, F_V_/F_M_ can be used as an indicator of photoinhibition, and NPQ quantifies non-photochemical energy dissipation [93].

A marked decline in the ratio of F_V_/F_M_ was observed in control plants following exposure to SO_2_ stress, while in plants treated with GABA, melatonin, and GABA + melatonin, the ratio of F_V_/F_M_ was higher than that of the control, which further explains the reduction in negative correlation between initial PI_abs_ and F_V_/F_M_ of plants primed with GABA compared to the control. In agreement with the results obtained from an experiments on *S. oblate* [42], our results also indicate that the capability of plants to use photon energy was not impaired in plants primed with melatonin or GABA + melatonin when exposed to SO_2_ stress. The same results were obtained by application of melatonin on tomato seedlings that were exposed to sulfuric acid-simulated acid rain, which depicted that melatonin treatment alleviated the acid-rain stress injury by increasing photosynthetic capacity [96]. Under SO_2_ stress, the NPQ was higher in primed plants compared to the control. In plants treated with GABA, the qT component of NPQ was higher than qE and equal to qI. In plants treated with melatonin, qE mainly contributed in improving NPQ, suggesting that GABA favors energy dissipation through state transition more than energy quenching through xanthophyll cycle, while melatonin improves energy quenching more than the other NPQ components under SO_2_ stress.

The higher correlation between PI_abs_ and NPQ in primed plants is due to higher initial PI_abs_ and NPQ compared with control, suggesting the conclusion that primed plants characterized with higher PI_abs_ dissipate the excess energy better than plants with lower photosynthetic efficiency, confirmed by improved F_V_/F_M_ in primed plants.

Osmotic stress is one of the most important abiotic stresses limiting plant growth [32] through dehydration; this restricts photosynthesis by pigment degradation and exerts irreversible damage to photosynthetic apparatus [97]. F_V_/F_M_ drastically decreased in leaf samples of control plants exposed to PEG, while sole application of GABA and melatonin or their co-application increased F_V_/F_M_ to obviously higher levels than non-primed plants under PEG stress. In agreement, in cucumber and soybean seedlings, osmotic stress significantly affected photosynthesis by decreasing F_V_/F_M_, and melatonin minimized the adverse effects of PEG [32,98].

PI_abs_ is one of the most sensitive parameters of JIP-test, which is considered as a generic response suitable to quantify responses to various abiotic stresses, such as drought, osmotic stress, salinity, air pollutants, etc. [99]. Under PEG stress, F_V_/F_M_ was negatively correlated with initial PI_abs_ (−98%), which may be explained by severe reduction of F_V_/F_M_ in response to PEG. GABA reduced this negative correlation to −19%, and melatonin and GABA + melatonin made the correlation positive and reached to 57% and 31%, respectively, which is due to the effect of GABA and melatonin in maintaining F_V_/F_M_ under PEG stress. The same finding has been reported by [32].

NPQ is a photoprotective mechanism acting through a set of biochemical processes that control light harvesting [100]. NPQ reflects the quenching of thermal energy through the xanthophyll cycle and other energy quenching pathways that are induced by the concentration of proton inside thylakoids [101]. Thermal energy dissipation at PSII was estimated by investigating the NPQ in leaf samples of plants treated with either or both melatonin and GABA. NPQ was increased when exposed to PEG stress, indicating that GABA and melatonin directed the excess excitation energy through non-radiative dissipative processes, which agrees with the results of previous study on the role of melatonin in alleviating PEG stress [32].

The correlation between initial PI_abs_ with NPQ was smaller and negative in control compared with plants primed with GABA, melatonin, and GABA + melatonin in both presence and absence of PEG stress. It can be concluded that the stronger correlation in primed plants compared with the control was due to higher initial PI_abs_ parallel with increase in NPQ under PEG stress. In this regard, GABA has been reported to enhance PEG-induced stress tolerance in *Piper nigrum* through increasing PSII activity [102], and melatonin showed to improve PI_abs_ in oat plants exposed to PEG stress, which minimizes the inhibitory effects of PEG [78].

Salt stress adversely affects crop productivity and survival by exerting osmotic and ionic stresses on plants [103]. GABA accumulation is an endogenous response of plants to salinity stress [28,104], suggesting a role for GABA in alleviating salt stress [20]. Our results showed that priming with GABA and melatonin minimized the adverse effect of NaCl by preventing the reduction of F_V_/F_M_. These findings agrees with those of the earlier studies, where enhanced F_V_/F_M_ was detected in GABA-treated plants under salt stress condition [104].

Melatonin improves the efficiency of light energy absorption and electron transport in PSII during salt stress [22]. The protective effect of melatonin on photosynthesis is due to alleviation of negative impact of NaCl on F_V_/F_M_ [29], which is in line with our results, where melatonin priming maintained F_V_/F_M_ in leaf samples that were exposed to NaCl stress. The same improving effect of melatonin has been reported in *Avena sativa* [105] and grapevine [106] plants. Considering the correlation between PI_abs_ and F_V_/F_M_ under NaCl stress, priming with GABA and melatonin increased the correlation compared to the control, which can be seen by higher F_V_/F_M_ in primed plants compared to control. Further, it is indicative of the protective role of GABA and melatonin on photosynthesis machinery, which leads to improved efficiency of the light reaction process in plants exposed to salinity stress.

When leaf discs of primed plants were exposed to NaCl, the value of NPQ was higher in plants primed with GABA and melatonin compared to control. The similar result has been reported when wheat plants were exposed to NaCl and GABA [23]. However, NPQ in leaf samples primed with melatonin was initially lower than the control before exposure to NaCl stress. The higher NPQ in NaCl-exposed samples indicated that the excitation energy was higher than the capacity of electron transport; as a result, it dissipates as heat that enables plants to withstand stressful condition [107]. Among NPQ components, qT and qI was higher than qE in plants primed with melatonin while in GABA-primed plants qE and qT was of the same level and higher than qI, suggesting that melatonin orients energy dissipation through state transition and favors the excitation energy distribution through PSII and PSΙ. At the same time, GABA causes the same level of energy dissipation through state transition and xanthophyll cycle.

The correlation between PI_abs_ before stress with NPQ after stress was negative in control plants, while in primed plants, this correlation was positive, indicating higher capacity of primed plants for dissipating excess energy, resulting in maintenance of PI_abs_ under stress condition by priming treatments. The protective role of melatonin on PI_abs_ under salt stress has been also reported in *Avena sativa* [105].

## 5. Conclusions

In the present study, priming with GABA and melatonin improved growth, stomatal conductance, and photosynthetic performance of *Vicia faba*. Priming caused higher initial photosynthetic performance of plants before being exposed to stressors, as indicated by higher initial PI_abs_. GABA and melatonin conferred to better photosynthetic performance under different abiotic stresses. Considering higher NPQ in primed plants when exposed to stresses, a role for priming agents in improving energy dissipation as a protective mechanism is conclusive. Not only was the quantity of energy dissipation affected by treatments but also the process of energy dissipation affected by both priming agents and stressors. The findings of the present study suggest a synergistic effect of GABA and melatonin since the growth parameters and photosynthetic performances was higher in plants primed with both GABA and melatonin compared to sole application of them. The use of chemical agents for priming to enhance plant tolerance against diverse abiotic stresses and discovering the underlying mechanisms of the action of priming substances is promising. The findings of the present study could provide a prompt approach for crop stress management as an alternative for other tedious techniques of breeding for improving stress tolerance of plants.

## Figures and Tables

**Figure 1 cells-10-01631-f001:**
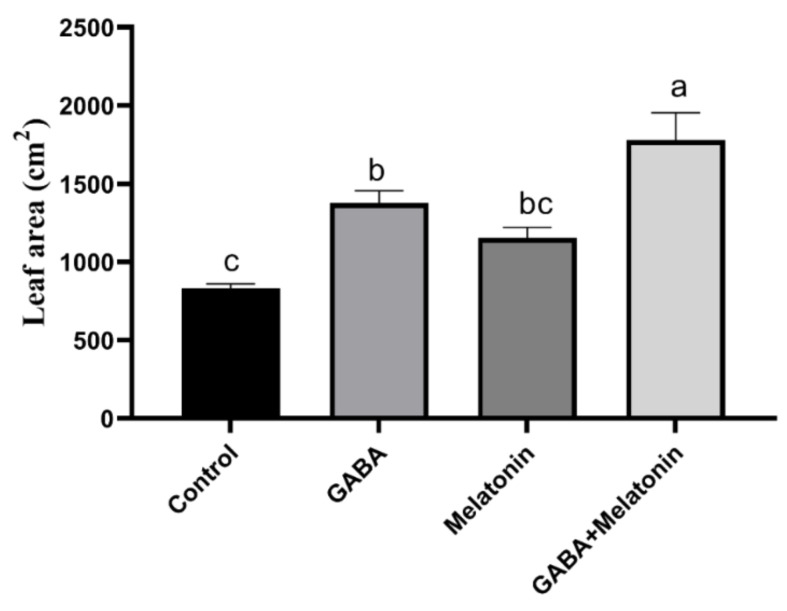
Effect of gamma-aminobutyric acid (GABA (20 µM)), melatonin (200 µM), and GABA (20 µM) + melatonin (200 µM) on leaf area of *Vicia faba.* Different letters denote significant differences among different treatments. Means followed by the same letter are not significantly different at 5% probability level by LSD test.

**Figure 2 cells-10-01631-f002:**
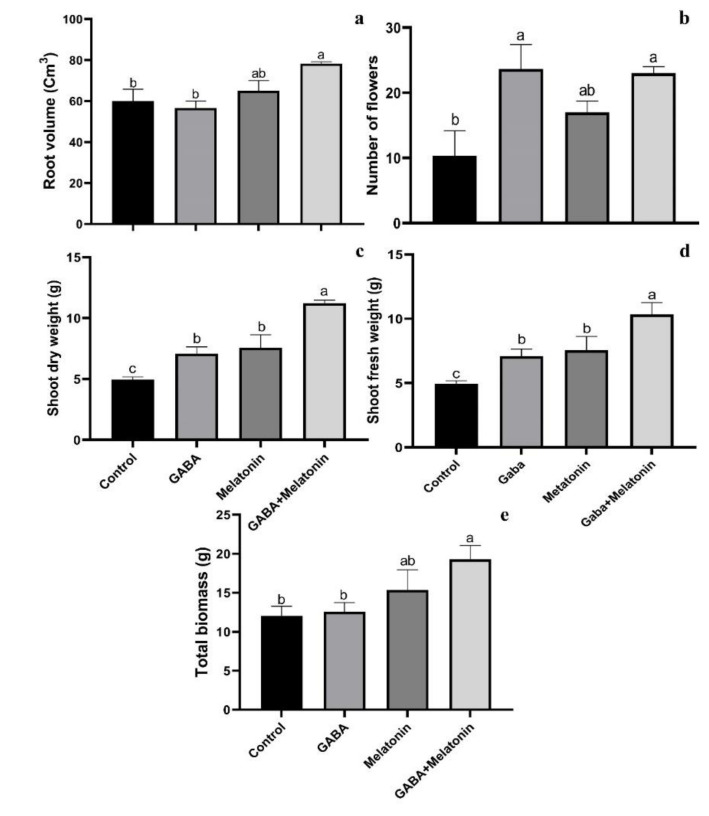
Effect of gamma-aminobutyric acid (GABA (20 µM)), melatonin (200 µM), and GABA (20 µM) + melatonin (200 µM) on root volume (**a**), number of flowers (**b**), shoot dry weight (**c**), shoot fresh weight (**d**), and total biomass (**e**) of *Vicia faba*.

**Figure 3 cells-10-01631-f003:**
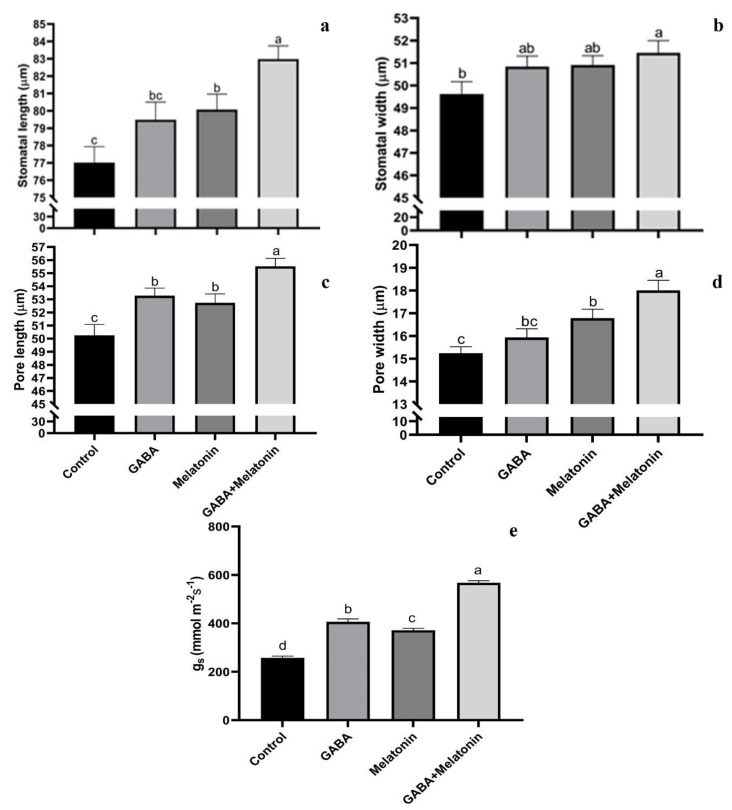
Effect of gamma-aminobutyric acid (GABA (20 µM)), melatonin (200 µM), and GABA (20 µM) + melatonin (200 µM) on stomatal width (**a**), stomatal length (**b**), length of stomatal pore (**c**), width of stomatal pore (**d**), and stomatal conductance (**e**) of *Vicia faba.* Different letters denote significant differences among different treatments. Means followed by the same letter are not significantly different at 5% probability level by LSD test.

**Figure 4 cells-10-01631-f004:**
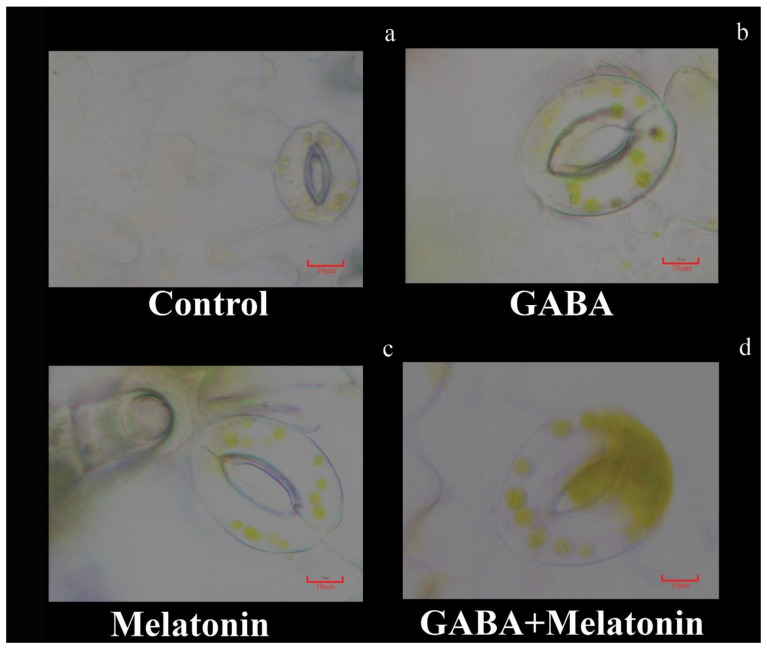
Effect of gamma-aminobutyric acid (GABA (20 µM)), melatonin (200 µM), and GABA (20 µM) + melatonin (200 µM) on stomatal characteristics of *Vicia faba.* Bar: (**a**–**d**): 10 µm.

**Figure 5 cells-10-01631-f005:**
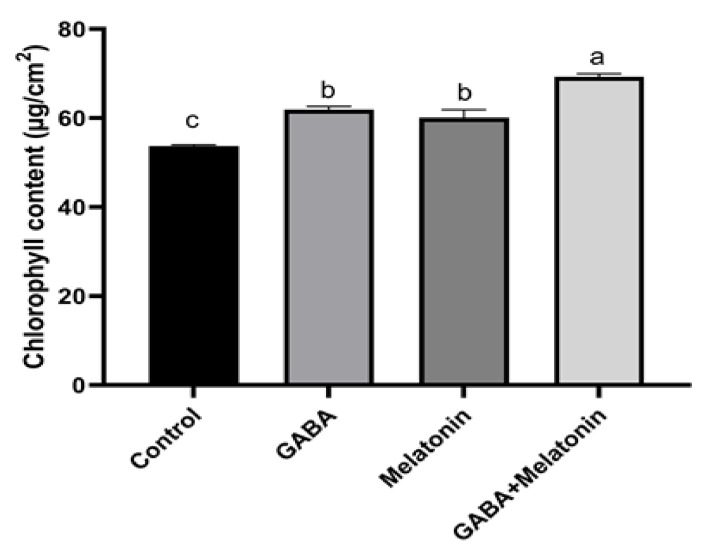
Effect of gamma-aminobutyric acid (GABA (20 µM)), melatonin (200 µM), and GABA (20 µM) + melatonin (200 µM) on chlorophyll content of *Vicia faba*. Different letters denote significant differences among different treatments. Means followed by the same letter are not significantly different at 5% probability level by LSD test.

**Figure 6 cells-10-01631-f006:**
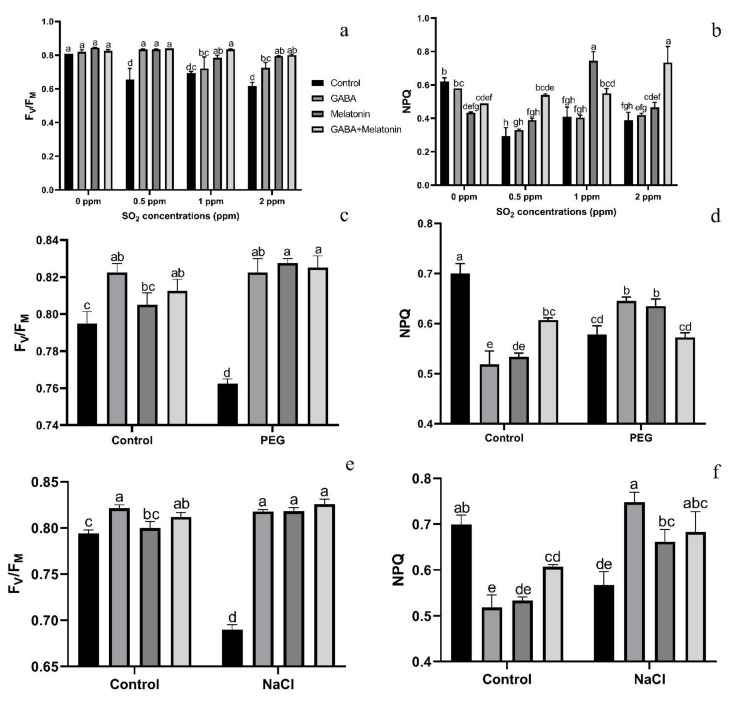
Effects of SO_2_, PEG, and NaCl stress on F_V_/F_M_ (**a**,**c**,**e**) and NPQ (**b**,**d**,**f**) in *Vicia faba* plants primed with gamma-aminobutyric acid (GABA (20 µM)), melatonin (200 µM), and GABA (20 µM) + melatonin (200 µM) and control. Different letters denote significant differences among different treatments. Means followed by the same letter are not significantly different at 5% probability level by LSD test.

**Figure 7 cells-10-01631-f007:**
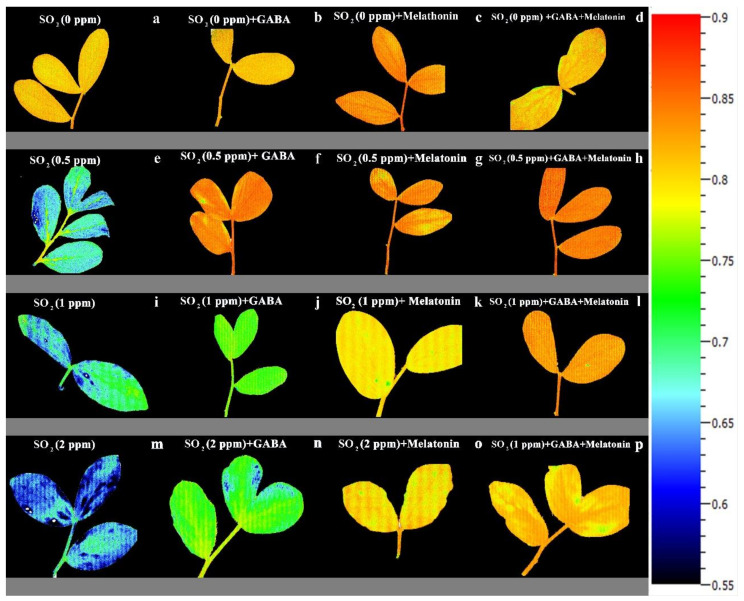
Effect of different concentrations of SO_2_ (0, 0.5, 1, 2 ppm) on F_V_/F_M_ of non-primed plants (**a**,**e**,**i**,**m**) and plants primed with gamma-aminobutyric acid (GABA (20 µM)) (**b**,**f**,**j**,**n**), melatonin (200 µM) (**c**,**g**,**k**,**o**), and GABA (20 µM) + melatonin (200 µM) (**d**,**h**,**l**,**p**) in *Vicia faba*.

**Figure 8 cells-10-01631-f008:**
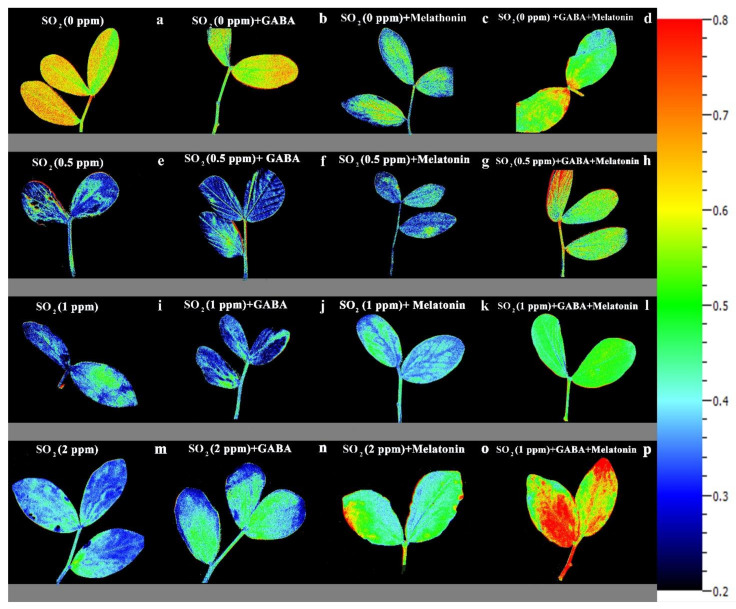
Effects of different concentrations of SO_2_ (0, 0.5, 1, 2 ppm) on NPQ of non-primed plants (**a**,**e**,**i**,**m**) and plants primed with gamma-aminobutyric acid (GABA (20 µM)) (**b**,**f**,**j**,**n**), melatonin (200 µM) (**c**,**g**,**k**,**o**), and GABA (20 µM) + melatonin (200 µM) (**d**,**h**,**l**,**p**) in *Vicia faba*.

**Figure 9 cells-10-01631-f009:**
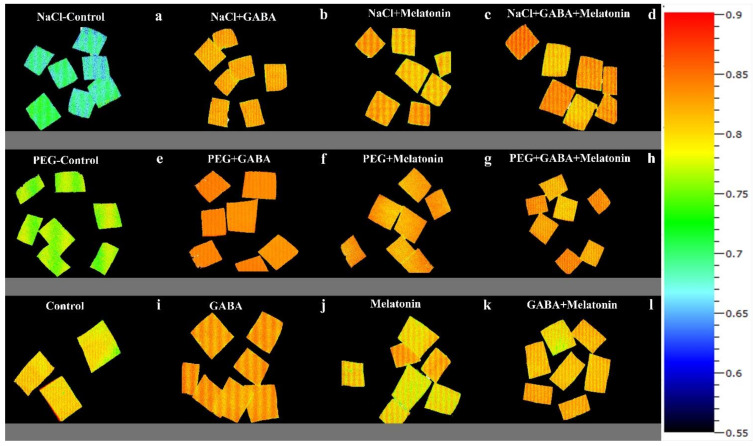
F_V_/F_M_ of *Vicia faba* plants primed with gamma-aminobutyric acid, (GABA (20 µM)), melatonin (200 µM), and GABA (20 µM) + melatonin (200 µM), and control following exposure to NaCl (0 and 100 mM) (**a**–**d**) and PEG8000 (0 and −8 bar) (**e**–**h**) and control (**i**–**l**).

**Figure 10 cells-10-01631-f010:**
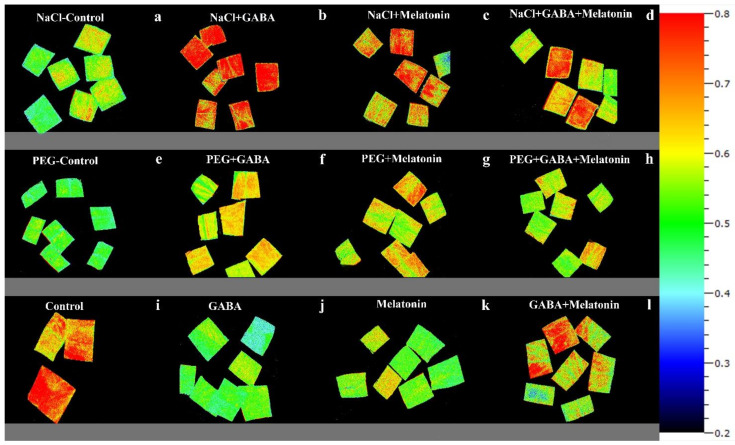
NPQ in *Vicia faba* plants primed with gamma-aminobutyric acid (GABA (20 µM)), melatonin (200 µM), and GABA (20 µM) + melatonin (200 µM), and control following exposure to NaCl (0 and 100 mM) (**a**–**d**) and PEG8000 (0 and −8 bar) (**e**–**h**) and control (**i**–**l**).

**Figure 11 cells-10-01631-f011:**
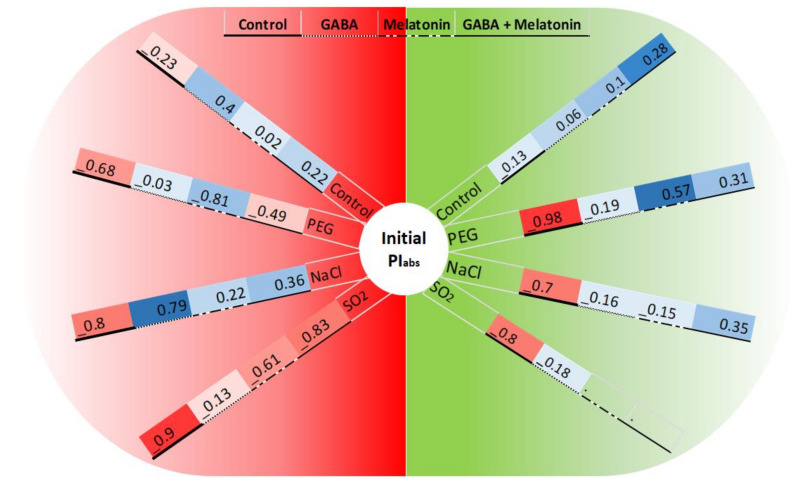
Changes in the correlation between initial PI_abs_ (before exposure to stressors) and F_V_/F_M_ or NPQ (the green and red semicircles related to the correlation between PI_abs_ and F_V_/F_M_ or NPQ, respectively) following exposure to PEG8000 (−8 bar), NaCl (100 mM), and SO_2_ (2 ppm) stresses and control. Blue color represents positive correlation, whereas red color represents negative correlation. Color intensity and size of the circle are proportional to the correlation coefficients.

**Table 1 cells-10-01631-t001:** Effect of application of gamma-aminobutyric acid (GABA (20 µM)), melatonin (200 µM), and GABA (20 µM) + melatonin (200 µM) on parameters obtained from OJIP transients of *Vicia faba*.

	F_V_/F_M_	PI_abs_	ABS/RC	TR_o_/RC	ET_o_/RC	DI_o_/RC
Control	0.77 ^c^ ± 0.008	1.05 ^c^ ± 0.24	2.95 ^a^ ± 0.086	2.30 ^a^ ± 0.058	1.05 ^b^ ± 0.034	0.65 ^a^ ± 0.030
GABA	0.80 ^ab^ ± 0.003	2.11 ^ab^ ± 0.11	2.55 ^b^ ± 0.084	2.06 ^b^ ± 0.060	1.15 ^a^ ± 0.053	0.49 ^b^ ± 0.024
Melatonin	0.80 ^b^ ± 0.006	1.93 ^b^ ± 0.26	2.56 ^b^ ± 0.036	2.06 ^b^± 0.022	1.12 ^ab^ ± 0.006	0.49 ^b^ ± 0.013
GABA+Melatonin	0.81 ^a^ ± 0.006	2.40 ^a^ ± 0.11	2.54 ^b^ ± 0.036	2.08 ^b^ ± 0.023	1.19 ^a^ ± 0.004	0.46 ^b^ ± 0.014

Different letters denote significant differences among different treatments. Means followed by the same letter are not significantly different at 5% probability level by LSD test.

## Data Availability

The datasets generated during and/or analyzed during the current study are available from the corresponding author on reasonable request.

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
