# Peer review of "Synergistic Effects of Melatonin and Gamma-Aminobutyric Acid on Protection of Photosynthesis System in Response to Multiple Abiotic Stressors"

_cells, 2021, doi:10.3390/cells10071631_

Round 1
Reviewer 1 Report
Shomali et al. presents results on synergistic, stress-protective effects of GABA and melatonin treatments on faba bean plants. This is a nice physiological work, supported by large amount of mainly chlorophyll fluorescence induction data as well as microscopy images. Such priming treatments may represent a useful, GMO-free way of providing stress tolerance to our crop plants, as the Authors also noted. While both GABA and melatonin treatments were already successfully applied for numerous plant species, I did not see many reports on them in case of faba bean and neither about their combination. In this sense, there is merit and novelty in the work.
However some of the results seemed surprising and incomplete to me. Based on growth and physiological parameters, priming with the above compounds managed to improve the general physiological status of the plants even under control conditions. It can be anticipated and it was proven too that these more healthy plants tolerated SO2, a toxic substance better. However the Authors also presented data and images showing that the treated plants had larger stomata with greater aperture. A lot of questions may arise at this point. Did the GABA and/or melatonin-treated plants have higher stomatal conductance compared to the control? How was their transpiration rate? How could they have higher osmotic stress tolerance (or had they)? Measurement of gas exchange parameters would be useful at this step (using e.g. LI-COR 6400 instrument). Providing at least basic growth parameters like fresh weight and representative images would also be useful. Further, I would dissuade the Authors from referring to the PEG treatment as drought. No, it is an osmotic treatment. There was no drought as far as I understood, the plants were kept in hydroponics.
There is some mistyping in the text, please correct them.
Abstract: please correct the writing of Vicia faba and also that of PIabs
MMs, page 3: modified Hoagland solution – how was it modified? Was it full strength?
MMs, page 4: calculation of Fv/Fm and NPQ can be omitted, well known to the readers
MMs, page 5: gas calculations must be considerably shortened, many of these laws are also known to readers
Results, page 6: ‘The maximum number of flowers observed in G and GM by respectively 56% and 55% increase in comparison with the control.’ – please change to increased
‘Plants treated with GM also showed the maximum total biomass by 37% increase in comparison with the control. Accumulation of biomass in plants treated with either of G or M was not significantly different with the control (Fig 2D).’ Please change as follows ‘Plants treated with GM also showed at maximum 37% total biomass increase, compared to the control. Accumulation of biomass in plants treated with either of G or M was not significantly different from the control (Fig 2D).’
Images, page 16: please write Fv/Fm as y axis label on Fig. 7 A, C, E in line with the text
Discussion, page 20: characeae, maiz – please correct
Page 21 vicia faba – please correct
Page 22 ‘In this research, both G and M increased in Faba bean.’ – I believe the Fv/Fm value has increased as result of these treatments
Page 24 pepper nigrum – please correct
Conclusions, Page 25: ‘The effects of G and M on plant growth and abiotic stresses tolerance studied previously.’ Please correct, I guess were or have been studied
‘resukts’ please correct
‘based on stress tolerant in plants’ please delete ‘in’
Author Response
Dear Reviewer,
Herewith, we submit the revision of the manuscript entitled “Synergistic Effects of Melatonin and GABA Gamma-Aminobutyric Acid (GABA) on Tolerance Induction to Multiple Abiotic Stressors” (Manuscript ID cells-1205899) by Aida Shomali, Sasan Aliniaeifard, Fardad Didaran, Mohammad Mohammadian, Mehdi Seif, WacÅ‚aw Roman Strobel and Hazem M. Kalaji for publication in Cell.
We would like to thank you for the valuable suggestions. Please find below a description on how we have dealt with the received comments. We hope that the manuscript can now be accepted for publication in your journal.
Sincerely,

Reviewer 2 Report
A brief summary
In the present work the effect of melatonin (M), gamma-aminobutyric acid (G), and their pair (MG) on the physiological parameters including PSII function was studied in faba bean plants. Additionally, the authors described the effect of M, G, MG on the physiological parameters of the plants under the action of some abiotic stresses.
Broad comments
The manuscript is written carelessly, some tables and figures duplicate each other, figure captions are written in short and do not contain anything about the experimental conditions and explanations of each panel, and some figures contradict the text.
In my opinion, the manuscript needs substantial revision, and figures and tables need to be brought to a serious scientific look.
Below I provide a list of my comments.
Probably, it is possible to write GABA in the main text, but I think that the full name should be written in the title.
In addition, the same should be in the abstract.
In the abstract, the authors indicate M and G concentrations (20 and 200 mkM), but it raises the question – Why these concentrations were chosen? I would recommend removing this from the abstract, but the reason for choosing these concentrations of G and M for the experiments should be clearly explained in the Introduction or in the Results. This information is absent in the presented text.
The first paragraph in the introduction is unnecessary in my opinion. The rest text of the introduction should be rewritten to present more clearly the known information about the effect of M and G on plants physiology, including photosynthesis (as well as PSII), the action of different concentration of M and G, the effect of such treatment on the closest relatives of faba bean plants as well as on other higher plants.
On the other hand, I propose to reduce a part about abiotic stresses.
In addition, the novelty of the study as well as the aim of the work should be written more clearly.
In the Methods the paragraph about “Application of abiotic stresses” (from Line 166) is written in a very heavy and confusing way. Rewrite it more clearly.
Many published articles contain the description of parameters calculated from OJIP curves. Therefore I recommend replacing table 1 in supplementary or just cite one of the previous work.
In the Results the authors write that “The maximum stomatal length was obtained in plants treated with GM; showing 7% longer stomata than the control…” (Line 245 and so on), but according to Fig. 4 the smallest stomata are in the case of GABA+Melatonin, if scale bars are the same in all panels. Unfortunately, the scale of the bars is not shown in the caption. How can the authors explain this discrepancy?
Through the text the order of experiments was G→M→GM. Why the authors changed this order in p. 3.3 and figs. 1 as well as table.2, fig 4, 5 8, 9 and so on, after that it was changed back: fig. 2, 3, 7 and so on. I would recommend correct it for the same format.
Table 2 and Fig 6 show the same results. I would recommend moving Fig 6 in Supplementary.
Why the authors have chosen SO2 as a stress factor? The information should be added.
Captions to almost all figs should be rewritten and after that include all needed information about conditions of experiments, scale bars, and so on.
The X-axes often do not contain units. For example, is the area of leaves in Fig 1 in cm, mm, or m? Fig 2 – is a volume of roots in m3? Fig 5 – is Chl content in g/L, mM?
As I mentioned above, Fig 4 does not correlate to the text. Scales of bars should be added.
Fig 7 is very difficult to understand. What does it mean Qy-Max? I could not find it in the text. And what about improving the caption?
The first part of the beginning of the Discussion probably for the Introduction. The Discussion should begin from the words… “Our data revealed that…” (Line 461).
Line 475. Why did the authors write about leaf area in this place?
Line 551. “In this research, both G and M increased in Faba bean.” – I propose that something lost here.
The paragraph in Lines 568–571 is unnecessary.
Line 585. “NPQ quantifies thermal dissipation” – What about state transitions (qT), PsbS protein (qE (together with Zea)), and photoinhibition (qI). All of them are parts of NPQ, but not all result in thermal dissipation. The same in Line 605. Can the authors present original induction curves of Chl fluorescence that they used for NPQ calculation in Supplementary?
Specific comments
Line 12. GABA – explain the abbreviation.
Line 15. “…GM (20 μM G +200 μM).” – Correct it.
Line 36. I think that “the physiological levels” could not be compared to “the molecular, cellular levels”. – Correct it.
Line 96. “G (50 μM /L), M (200 μM) and G (50 μM) + 96 M (200 μM)” – Why L is only in the first concentration? Correct it.
Line 140. “…details). was calculated using a custom-made…” – What was calculated?
Line 159. This is only one moment with a Chl abbreviation. Correct it.
Line 160. Here and below: OJIP test →JIP-test (10.1080/02827581.2010.485777).
Line 174. The ideal gas low is absent in the presented text.
Line 672. “the resukts suggests” – Correct it.
Author Response

(The authors gave the same response as above.)

Round 2
Reviewer 1 Report
The text has improved considerably (though please check the end of the Introduction, some parts seem to be repeated). I found some, but not all of my concerns addressed. So as I see, it is a very nice and interesting physiological and cell biological piece of work. That is why I would encourage the Authors do some more experiments, to make it more complete. So the Authors successfully tried the priming effect of GABA and melatonin on hydroponically grown plants, but somehow (why?) shifted to the treatment of leaf pieces thereafter. This way they could prove the protection of the photosynthetic machinery under these stress conditions, but could not prove the better stress tolerance of the entire plant. So to my opinion, the MS might be published as it is now, in a minor journal with more modest title and more modest conclusions. However it cannot be published in a 4 IF journal without proving the stress tolerance of entire plants in hydroponics. I encourage the Authors to carry out these experiments (possibly just a few weeks) and measure at least basic parameters like fresh weight. My concern is still that plants with greater stomatal conductance would transpirate more and have less osmotic stress tolerance. If the Authors want to publish the opposite, they have to prove it.
Author Response
Dear reviewer,
We would like to thank you for the kind words and the valuable suggestions. Please find in attachment a description on how we have dealt with the received comments.
Yours sincerely,

Reviewer 2 Report
Broad comments
Surprisingly, in the new version the text of the manuscript is still carelessly written. See my comments below. I would like to recommend that the authors take the preparation of the text for publication more seriously. They should be more attentive. It is obvious that the presented version can not be published.
Lines 2-3. The abbreviation in the title is not allowed. Delete it.
In the previous version of the manuscript, the concentration of used GABA was indicated as 20 mkM (Line 17 (deleted)). In the present version this is changed to 50 mM. I.e. the difference in values is about 2500 times! At the same time, in Line 85 the authors write about 25-50 mkM of GABA used by other authors (Seifikalhor et al., 2020). I think that such changes in the indication of used concentrations in the submitted manuscript are unacceptable if there is no obvious reason for it. Therefore, the authors should have written why they made such changes. Unfortunately I can not find this in the cover letter. In addition, the authors should have explained the use of such a high concentration of GABA as compared to other studies (Seifikalhor et al., 2020).
Paragraph between lines 124 and 144. The paragraph was rewritten. However, now it is written very confusedly. Correct it.
Through the text the order of experiments was G→M→GM. Authors should use this order through the entire text including Table 1. I indicated this in the previous revision. Be attentive! For example: Table 1, Lines 117, 118, 151-152, 291, 294, 331 and so on. Check the entire text, please.
I think that equations used for Fv/Fm and NPQ calculations should be presented in the Methods. (Lines 223 and 228).
Line 230. OJIP test measurements → OJIP test measurements or OJIP test measurements
Fig. 4. The bar size should be indicated in the caption.
Fig. 6. It will be useful if induction curves used for Fv/Fm and NPQ calculations were arranged in the same order in the Supplementary. In addition, it is not necessary to write Fv/Fm and NPQ in full in the caption (The same in Fig 7-10).
In my previous reply, I asked to remove a paragraph between lines 683 and 686. The authors deleted a paragraph between lines 674 and 678, in spite of the fact that this text is very useful in my opinion.
From line 720. In my first reply, I asked the authors about the addition of information concerning other participants of NPQ in higher plants besides carotenoids. So, I repeat my question: “What about state transitions (qT), PsbS protein (qE (together with Zea)), and photoinhibition (qI). All of them are parts of NPQ, but not all result in thermal dissipation?” Please, add this information to the text.
The paragraph from line 804. Correct the description of the Supplementary according to Tables and Figures presented there now. It’s incorrect in the present text.
Specific comments
Line 74. GABA and Melatonin → GABA and Mmelatonin
Line 78 and below. The authors should choose whether to use photosystem II or PSII in the text.
Line 81. Non photochemical quenching (NPQ) → Non photochemical quenching of chlorophyll fluorescence
Lines 95, 97. by maltonin – correct it.
Line 97. were chractrized – correct it.
Line 117. M → melatonin
Line 130 – M and G. Correct it.
Author Response

(The authors gave the same response as above.)

Round 3
Reviewer 1 Report
While the authors did not provide the proposed results from hydroponic experiments, they provided strong arguments to their cause. OK, I accept that there is no mechanistic relationship between osmotic/drought stress tolerance and stomatal conductance/transpiration. I see that melatonin treatment increased the drought tolerance of other plants despite also increasing the stomatal conductance. I think the subject was worth of a discussion and I am happy that the arguments become part of the MS.
Author Response
Dear Reviewer,
We appreciate your valuable comments and remarks about our manuscript which improved it significantly. We did our best to add required information to the manuscript according to your comments.
Regards
Reviewer 2 Report
Broad comments
I appreciate the significant work that the authors did with the text of the manuscript.
I can even say, that I recommend the manuscript for publication if based only on the science part of the manuscript. At the same time, I am surprised (again) that the authors are still inattentive to their text. Bellow, I indicate some incorrect points in the manuscript, but the authors should read the entire text, probably not once, but several times, and correct ALL the incorrect moments.
Line 26. … chlorophyll fluorescence. As in Line 84.
Line 28. FV/FM→ FV/FM
Lines 305-306. “…treated with GABA, GABA + melatonin and melatonin respectively showed 39%, 113%, and 38%...”. I disagree that the “…the treatments showing highest change to lowest changes…” is correct here as the authors replied in a Cover letter.
Line 342. Probably the paragraph containing the description of Fig 3 C and D is lost (Lines 322–329 in the old version).
Line 344. Figure 1 →Figure 3.
Line 353. Surprisingly, but Fig. 4 is not mentioned in the main text, but should to.
Line 363. Figure 2 →Figure 5.
Line 366. There is no Table 2. Table 2 → Table 1
Line 735. I don’t understand why the authors refuse to describe other NPQ components besides qZ, nevertheless, they should do it in this place.
Line 825. OJIP test → OJIP-test.
Line 830. Figure S3: 3 NPQ Partitioning kinetics → Figure S3: 3
Lines 831-832. “…when exposed to NaCl stress (100 mM) (B), PEG stress (-8 bar) (C) and SO2 stress (2 ppm) (D)”. This not correlates with Figure S3. There is another order there. Correct.
Lines. 836, 841, and 845. Correct numbering of related figures in SM.
SM Figure S3 (and Line 427). Correct the order of qE, qT, and qI in the description.
Author Response
Dear Reviewer,
We appreciate your valuable comments and remarks about our manuscript which improved it significantly.
In fact some of the criticism is due to sending an incomplete version, which is a bit different from the revised submitted manuscript to the Cells. We don't know why this happened. Hopefully this time the correct version will be sent to the reviewers. We also read the manuscript several times to remove any punctuation and wording problems of the manuscript.
We did our best to do required corrections according to your comments.
Regards
